# Antioxidant, Anti-Inflammatory, Anti-Menopausal, and Anti-Cancer Effects of Lignans and Their Metabolites

**DOI:** 10.3390/ijms232415482

**Published:** 2022-12-07

**Authors:** Won Young Jang, Mi-Yeon Kim, Jae Youl Cho

**Affiliations:** 1Department of Integrative Biotechnology, Sungkyunkwan University, Suwon 16419, Republic of Korea; 2School of Systems Biomedical Science, Soongsil University, Seoul 06978, Republic of Korea

**Keywords:** lignans, antioxidant, chronic inflammation, menopausal symptoms, cancers

## Abstract

Since chronic inflammation can be seen in severe, long-lasting diseases such as cancer, there is a high demand for effective methods to modulate inflammatory responses. Among many therapeutic candidates, lignans, absorbed from various plant sources, represent a type of phytoestrogen classified into secoisolariciresionol (Seco), pinoresinol (Pino), matairesinol (Mat), medioresinol (Med), sesamin (Ses), syringaresinol (Syr), and lariciresinol (Lari). Lignans consumed by humans can be further modified into END or ENL by the activities of gut microbiota. Lignans are known to exert antioxidant and anti-inflammatory activities, together with activity in estrogen receptor-dependent pathways. Lignans may have therapeutic potential for postmenopausal symptoms, including cardiovascular disease, osteoporosis, and psychological disorders. Moreover, the antitumor efficacy of lignans has been demonstrated in various cancer cell lines, including hormone-dependent breast cancer and prostate cancer, as well as colorectal cancer. Interestingly, the molecular mechanisms of lignans in these diseases involve the inhibition of inflammatory signals, including the nuclear factor (NF)-κB pathway. Therefore, we summarize the recent in vitro and in vivo studies evaluating the biological effects of various lignans, focusing on their values as effective anti-inflammatory agents.

## 1. Introduction

As the name suggests, phytoestrogens are estrogen analogs. These can be extracted from flax seeds, sesame seeds, fruits, and vegetables [1]. They have polyphenolic structures and can bind to estrogen receptors in the human body, mimicking the effects of estrogen. Therefore, many reports have discussed the therapeutic potential of phytoestrogens for estrogen replacement. Phytoestrogens include isoflavonoids, flavonoids, stilbenoids, and lignans [2]. Lignans are classified into seven main types: secoisolariciresionol (Seco), pinoresinol (Pino), matairesinol (Mat), medioresinol (Med), sesamin (Ses), syringaresinol (Syr), and lariciresinol (Lari) (Figure 1) [3]. Many lignans exhibit therapeutic effects, including antioxidant, anti-cancer, anti-inflammation, anti-bacterial, and anti-fungal properties [4,5,6,7]. Most bioactivities of lignans arise following their chemical transformation by gut microbiota. For instance, undigested flax or sesame seeds contain a secoisolariciresionol diglucoside (SDG) conjugate [8]. The human gut microbiota is responsible for the conversion of SDG to Seco by deglycosylation. The unconjugated Seco can be demethylated or dehydroxylated by two fecal bacterial strains, *Peptostreptococcus productus* and *Eggerthella lenta*, to produce enterodiol (END) or enterolactone (ENL) metabolites that are easily absorbed from the gastrointestinal tract [9]. Similarly, microbial digestion of other lignans within the gastrointestinal tract may also lead to production of the END or ENL metabolites in the colon. 

Inflammatory processes are initiated by binding of pathogen-associated molecular patterns (PAMPs) to pattern recognition receptors (PRRs) [10,11,12]. Major PRRs include toll-like receptors (TLRs), which activate TGF-β-activated kinase 1 (TAK1) to induce the phosphorylation of downstream molecules of the nuclear factor (NF)-κB and activator protein (AP)-1 pathways [13]. NF-κB subunits, including Rel A (p65) and p50, bind to IκB inhibitory protein [14,15,16]. However, TAK1-mediated phosphorylation of IκB kinase (IKK) activates the kinase activity of IKK, inducing the proteasomal degradation of IκB. This triggers the release NF-κB subunits and phosphorylation of p50 and p65 [17,18]. TAK1 also increases phosphorylation of mitogen-activated protein kinase (MAPK), including p38, extracellular signal-regulated kinase (ERK), and c-Jun N-terminal kinase (JNK), resulting in the continuous phosphorylation of AP-1 subunits, e.g., c-Jun and c-Fos [19,20,21,22,23]. Both NF-κB and AP-1 translocate into the nucleus after their phosphorylation, acting as a pivotal transcription factor for inflammatory responses [24,25]. Immune cells utilize these processes to secrete pro-inflammatory cytokines that trigger neutrophil infiltration and activation of lymphocytes for further immunological activation [26,27]. However, chronic inflammation may result in damage to normal tissues, causing necrosis of non-infected cells [28,29,30]. Excessive inflammation also recruits reactive oxygen species (ROS) that may alter tissue metabolism [31,32]. In postmenopausal women suffering from estrogen deficiency, chronic inflammation can be significant in the reproductive organs, aorta, bone, brain, and other organs. Progression of most symptoms during the menopausal period is modulated by the NF-κB pathway [33,34]. Increased cytokine production induced by NF-κB pathway activation in various organs may eventually lead to the progression of atherosclerosis, osteoporosis, and psychological disorders such as depression [35,36,37,38,39,40]. In addition, it is also reported that NF-κB activation is known to trigger cancer cell proliferation and angiogenesis by regulating the gene expression of B-cell lymphoma 2 (Bcl-2), vascular endothelial growth factor (VEGF), and colony stimulating factor 1 (CSF1) [41,42]. Long-term secretion of pro-inflammatory cytokines, which can be induced by NF-κB and AP-1, may also activate various pathways of cell migration, apoptosis, and proliferation, associated with cancer survival [43]. For instance, elevated tumor necrosis factor alpha (TNF-α) and interleukin-6 (IL-6) secretion is known to promote the metastasis and malignancy of tumor cells [18,44,45,46]. Therefore, the development of novel methods to alleviate chronic inflammation remains a therapeutic target in many diseases [12].

In this review, we summarize recent studies of the therapeutic properties of various lignans, especially those targeting chronic inflammatory diseases (Figure 2). For this, we referred to the experimental papers published within the last five years that show the efficacy of lignans. We set various lignan compounds, including the seven classes and their metabolites, as key words and searched studies in Pubmed and Embase. To be specific, the efficacy of lignans in both in vitro and in vivo models of menopausal disorders and cancers areis highlighted in a discussion on the outlook of lignan research. 

## 2. Antioxidant Properties of Lignans

ROS, including free radicals such as the superoxide (O2*) or hydroxyl (HO*) radicals and non-radical molecules such as hydrogen peroxide (H2O2), are crucial factors for pathogen resistance and tumor removal, in that elevated ROS concentrations precede programmed cell death of pathogen-infected or cancer cells [47,48]. However, excessive or unfavorable production of ROS often leads to the progression of severe inflammatory diseases [31]. For instance, inflammatory bowel disease (IBD; characterized by ulcerative colitis) is associated with the uncontrolled release of cytotoxic ROS, which contributes to the increased expression of pro-inflammatory molecules, such as leukotriene B4 [49]. Therefore, many antioxidant reagents, e.g., ascorbic acid or riboflavin, have been examined as therapeutic agents in inflammatory diseases [50,51]. Interestingly, many experiments have demonstrated the antioxidant effects of lignans (Figure 3). Below, we summarize how each type of lignan ameliorates oxidative stress in vitro and in vivo.

It has been found that SDG, the glycosylated form of Seco, was able to upregulate the expression of superoxide dismutase (SOD) and catalase (CAT) in damaged mouse liver and kidney [52,53]. Since both enzymes scavenge ROS, we can assume that SDG induces free radical scavenging [54,55]. Moreover, glutathione concentrations in both the liver and kidney increased to normal due to SDG treatment, while malondialdehyde (MDA) concentrations were reduced. Ralph A. Pietrofesa et al. confirmed that synthetic SDG ameliorates ROS production through activation of nuclear factor erythroid 2-related factor 2 (Nrf2), which binds to the AU-rich element (ARE element) to induce the expression of genes for antioxidant proteins, such as heme oxygenase (HO-1) [56,57]. The Lari lignin has two enantiomer structures, (+)-lariciresinol and (−)-lariciresinol: the antioxidant activity of the (+)-lariciresinol enantiomer extracted from *Rubia philippinensis* increased Nrf2–induced HO-1 expression in RAW264.7 murine macrophage cells [58].

## 3. Anti-Inflammatory Properties of Lignans

Inflammatory responses play an important role in eliminating invasive pathogens during the early stages of infection. However, chronic inflammation accompanies many diseases including cancer [59,60,61,62,63]. Moreover, conventional anti-inflammatory agents, e.g., the corticosteroid dexamethasone, exert severe side effects, e.g., muscle atrophy [64], while nonsteroid anti-inflammatory drugs (NSAIDs) specifically target cyclooxygenases (COXs) [65]. Therefore, many studies addressing modulation of inflammatory signaling pathways by natural compounds have been conducted. As a result, lignans derived from many plants have been found to possess anti-inflammatory activities. 

Various molecular mechanisms contribute to pro-inflammatory processes. For instance, the Janus kinase (JAK)-signal transduction and activator of transcription (STAT) signaling pathway is related to T-cell differentiation and B-cell class switching by receiving signals from both type I and type II cytokines [66]. Meanwhile, the NF-κB and MAPK pathways can be activated by phosphorylation and translocation of NF-κB and AP-1 subunits, respectively [67,68]. Both transcription factors upregulate pro-inflammatory cytokines, including TNF-α, interleukin-1 beta (IL-1β), IL-6, COX-2, and inducible NO synthase (iNOS) [69,70,71]. Interestingly, it has been reported that dietary lignans and their metabolites derived from gut microbiota control the inflammatory response through suppression of these pathways [72,73]. 

Zhen Wang et al. explored whether SDG could attenuate intestinal inflammation in C57BL/6 mice [74]. The research group discovered that, in a dextran-sulfate-sodium-salt (DSS)-induced colitis model, mice supplemented with 200 mg/kg of SDG in their diets had reduced pathological severity. Administration of SDG blocked the NLR family pyrin domain containing 1 (NLRP1) inflammasome due to NF-κB pathway inhibition. Moreover, oral administration of SDG increased the serum enterolactone concentration and attenuated atopic dermatitis in a mouse model [75]. Increased enterolactone attenuated the Th2 cell response by targeting the JAK-STAT6 signaling pathway. The Pino-mediated anti-inflammatory response was dependent on NF-κB pathway inhibition accompanied by phosphorylation of upstream molecules like IκBα [76]. Meanwhile, Syr treatment attenuated the expression of inflammatory cytokines, such as TNF-α, IL-1β, COX-2, and iNOS in BV2 microglial cells [77]. These effects were induced by inhibition of translocation of the p65 subunit of NF-κB to the nucleus. Along with the effects of Syr, dose-dependent inhibitory effects of Ses on p65 phosphorylation accompanied the downregulation of TNF-α, IL-1β, and interleukine-8 (IL-8) secretion in an vivo carrageenan-induced model of inflammation [78]. Schisandrin B, the lignan found in *Schisandra chinensis*, also exerted anti-inflammatory effects by modulating both the NF-κB and MAPK pathways [79].

Taken together, the reported findings indicate lignans as potential therapeutic agents in inflammatory conditions (Figure 4).

## 4. Anti-Menopausal Effects of Lignans

The ovary plays a crucial role in the female reproductive, cardiovascular, skeletal, and central nervous systems in that it, along with the brain, is the major organ that secretes estrogen [80]. In general, estrogen secretion is mediated by luteinizing hormone (LH) and follicle-stimulating hormone (FSH), which are produced based on a signal from gonadotropin synthesized in the hypothalamus [81]. The process is regulated by negative feedback in that secreted estrogen downregulates the secretion of LH, FSH, and gonadotropin. However, estrogen deficiency due to ovarian aging blocks the negative feedback system, leading to various menopausal symptoms. Such symptoms include face flushing, skin dryness, and anxiety and may be accompanied by more severe conditions such as cardiovascular disorders, osteoporosis, and depression. Most symptoms are correlated with chronic inflammation as estrogen deficiency often leads to a dramatic increase in pro-inflammatory cytokine concentrations in serum, liver, bone, and brain. To alleviate postmenopausal symptoms, hormone therapies with direct injection of estrogen have been developed based on certain guidelines from the US FDA. However, treatment-related breast and uterine cancer has limited the use of direct estrogen administration as a mainstream treatment [82]. Numerous studies have been conducted to evaluate the potential of lignans as therapeutic agents for menopausal symptoms. 

### 4.1. Cardiovascular Disease

Most women in menopause suffer from a vasomotor system disorder [83]. Symptoms last several years in some cases and are accompanied by redness of the skin and night sweating. The symptoms are not life-threatening but significantly reduce the quality of life. 

Cardiovascular disease is less prevalent in premenopausal women than in men, but estrogen deficiency increases the prevalence of cardiovascular disease [84]. This change is a function of estrogen receptor alpha (ERα) and estrogen receptor beta (ERβ), which are activated by estrogen and phosphorylate endothelial nitric oxide synthase (eNOS) [85]. Activation of eNOS induces nitric oxide (NO) production, which mediates vascular relaxation. According to several studies, estrogen also increases serum high-density lipoprotein (HDL) and reduces serum low-density lipoprotein (LDL) [86]. Increased LDL level induces chronic injury to the aorta and liver by recruiting and activating macrophages to increase production of cytokines including TNF-α, IL-1β, and plasma platelet activating factor-acetyl hydrolase (PAF-AH) [87]. Since excessive LDL causes chronic inflammation in the aorta and liver, estrogen deficiency can lead to hypercholesterolemia and atherosclerosis. 

Hui-Hui Xiao et al. showed that the lignan-rich fraction from *Sambucus Williamsii* Hance attenuated the imbalance between HDL and LDL levels in an ovariectomized (OVX) mouse model, as well as reducing total cholesterol (TC) and triglyceride (TG) concentrations. The extract significantly reduced serum and liver LDL concentrations. The extract also reduced serum total cholesterol and triglycerides in OVX mice by reducing serum concentrations of various cytokines, including TNF-α, interleukin-22 (IL-22), and monocyte chemoattractant protein-1 (MCP-1), and modulating the gut microbiota population [88]. In light of the effects of lignans on menopause-related cardiovascular diseases, a clinical investigation of 214,108 people was undertaken, and long-term intake of lignans was found to significantly downregulate the risk of total coronary heart disease in both men and women [89]. Surprisingly, steady intake of 40 μg/day of Mat reduced the hazard ratio of coronary heart disease by almost half compared to the group that did not consume the lignan [89]. In addition, oral administration of 50 mg/kg of sesame lignans attenuated oxidized LDL in rabbits fed a fat- and cholesterol-enriched diet and reduced inflammatory lesions in liver induced by a high-fat diet [90]. Among the sesame lignans, Ses also alleviated cardiovascular injury in rats by reducing secretion of lactate dehydrogenase (LDH), creatine kinase (CK), and CK-MB in the heart and serum [91]. In addition, Ses reduced iron accumulation in the heart by recovering the proteins related to ferroptosis inhibition including solute carrier family 7 member 11 (SLC7A11) and glutathione peroxidase 4 (GPX4). The Ses lignin also activated the transient receptor potential vanilloid type 1 (TRPV1) that phosphorylates downstream molecules including protein kinase A (PKA), protein kinase B (Akt), and AMP-activated protein kinase (AMPK), which stimulate eNOS for NO secretion [92]. Interestingly, Ses-mediated eNOS activation inhibited the TNF-α-induced inflammatory response by downregulating NF-κB signaling and intracellular adhesion molecule-1 (ICAM-1) expression. This finding indicated that Ses may prove useful to prevent hypertension and coronary inflammation. Summary of the result of published pharmacological experiments conducted to evaluate the effects of lignans on cardiovascular diseases is included in Table 1.

### 4.2. Osteoporosis

Osteoporosis results from an imbalance between the activities of osteoblasts and osteoclasts [93] and is characterized by reduced bone mineral density and degradation of trabecular tissues, resulting in fragility fractures. 

Bone marrow stromal cells differentiate into osteoblasts in the presence of transcription factors, such as osterix (OSX), SRY-box9 (SOX9), and runt-related transcription factor 2 (RUNX2). Osteoblasts then secrete alkaline phosphatase (ALP), osteopontin (OPN), and osteocalcin (OCN) for bone formation. Bone marrow stem cells or bone marrow-derived macrophages differentiate into osteoclasts in the presence of macrophage-colony stimulating factor (M-CSF) and receptor activator of nuclear factor-κB ligand (RANKL) [94]. The interaction between RANK from osteoclasts and RANKL precedes a downstream signal along the MAPK or NF-κB pathway to stimulate nuclear factor of activated T cells 1 (NFATc1) [95,96]. Then, NFATc1-mediated expression of protease-like cathepsin K (CTSK) or matrix metalloproteinase-9 (MMP-9) triggers the degradation of the collagen structure in bone, which recruits C-telopeptide of collagen Type 1 (CTX-1) in blood [34,97]. The activity of osteoclasts can be blocked by the actions of estrogen, while estrogen upregulates the production of osteoprotegerin (OPG) in osteoblasts [98]. Since OPG is a decoy receptor that competitively binds to RANKL against RANK, estrogen-mediated OPG expression reduces bone resorption by inhibiting RANKL-induced differentiation of osteoclasts and further downstream signals [99]. Unfortunately, estrogen deficiency in menopausal women results in significant permanent bone loss and absence of osteoclast regulating factor.

Although other lignans such as Mat repress osteoclastogenic activity by targeting the MAPK pathway of osteoclasts, recent studies of the activity of lignans in osteoporosis have mainly focused on the effects of Ses [100]. When Ses was added to bone marrow stem cell cultures, it promoted gene expression of osteoblast differentiation markers like RUNX2 or osteocalcin. Moreover, Ses blocked protein expression of glycogen synthase kinase 3 beta (GSK3β), enhancing the wingless-related integration site (Wnt)/β-catenin signaling pathway to increase osteoblast proliferation [101]. Zhengmeng Yang et al. also discovered the anti-osteoclastogenesis effect of Ses to be mediated by NF-κB pathway inhibition, which reduces the expression of CTSK and tartaric-acid phosphatase (TRAP) in osteoclasts [102]. Moreover, healing of osteoporotic fractures in C57BL/6 mice following OVX surgery was improved by oral administration of Ses [103]. Ses upregulated the expression of VEGF, an angiogenesis marker, along with bone morphogenetic protein 2 (BMP2) and SOX9, both chondrogenesis markers. These effects promoted cartilage formation, supporting bone stability and the synthesis of trabecular bone tissues in OVX mice. After Ses, sesamolin is the second major lignan derived from sesame [104]. Intraperitoneal injection of sesamolin in OVX mice improved the volume and number of trabecular tissues in the femur [105] due to downregulation of phosphorylation of p65 and MAP kinases including ERK, JNK, and p38. These studies indicate a potentially significant role for lignans, and especially those derived from sesame seeds, as therapeutic agents for postmenopausal osteoporosis. Summary of the result of published pharmacological experiments conducted to evaluate the effects of lignans on osteoporosis is included in Table 1.

### 4.3. Psychological Disorders

Although the mechanism is not clearly understood, postmenopausal women often suffer from depression accompanied by irritability, fatigue, and loss of confidence. Recently, several effects of estrogen and its receptors have been identified in serotonergic and dopaminergic systems. Specifically, estrogen activates the tryptophan hydroxylase responsible for synthesis of serotonin (5-HT) [106]. Moreover, estrogen mediates the mRNA expression of monoamine oxidases (MAO), which act as serotonin degradation enzymes [107]. In the postmenopausal period, reduced 5-HT in the brain increases the feeling of anxiety, eventually leading to depression. Ovariectomy-induced estrogen deficiency increases neuroinflammation together with increased TNF-α and IL-1β levels in the neurons of the dorsal root ganglion, hippocampus, and spinal dorsal horn [108]. Neuroinflammation reduces magnesium ion concentrations in neurons, eventually leading to memory loss and chronic pain. 

When designing an animal model of depression, researchers often create chronic unpredictable mild stress (CUMS) among the test animals for several weeks [109]. Yihang Zhao et al. developed a CUMS protocol consisting of food/water deprivation, overnight illumination, forced swimming, cage tilting, and tail clipping. In mice subjected to these stressors for six weeks, 50 mg/kg/day of Ses upregulated 5-HT level and decreased norepinephrine level in the striatum [110]. Moreover, CUMS-induced loss of brain-derived neurotrophic factor (BDNF) expression was strongly attenuated by Ses administration. Qianxu Wang et al. also found that Ses-mediated anti-depressant effects are strongly correlated with its inhibitory effects on neuroinflammation [111]. Ses downregulated cytokine expression in the mouse cortex and protected the brain from inflammatory injuries. Summary of the result of published pharmacological experiments conducted to evaluate the effects of lignans on psychological disorders is included in Table 1.

**Table 1 ijms-23-15482-t001:** Summary of the result of published pharmacological experiments conducted to evaluate the effects of lignans on postmenopausal symptoms.

Disease	Lignan	Source	Test Type and Dose	Molecular Mechanism	Ref.
Cardiovascular	Lignan-rich Fraction	*S. Williamsii*	In vivo (140, 280 mg/kg)	Gut microbiota modulation TNF-α ↓ LDL, TG, TC ↓	[88]
	Total Lignan, Mat, Seco, Pino, Lari	Dietary Lignans	Clinical test (Diet was repeatedly assessed using questionnaire)	Circulating ENL↑ Total fiber intake ↑ Coronary head disease risk ↓	[89]
	Lignans (Sesamin:Episesamin = 1:1)	Purchased (Takemoto Oil & Fat)	In vivo (50 mg/kg)	LDL ↓ PAF-AH ↓ IL-1β, macrophage infiltration ↓	[90]
	Ses	Purchased (Aladdin)	In vivo (40, 80, 160 mg/kg)	LDH, CK, CK-MB ↓ TNF-α, IL-1β ↓ SOD, GSH ↑ MDA ↓ SLC7A11, GPX4 ↑	[91]
	Ses	Purchased (Sigma-Aldrich)	In vitro (20 μM)	TRPV1, PKA, Akt, AMPK ↑ eNOS ↑ p65, ICAM-1 ↓	[92]
Osteoporosis	Mat	Purchased (Sigma-Aldrich)	In vitro (10 μM)	TRAP ↓ NFATc1 ↓ p38, ERK ↓	[100]
	Ses	Purchased (Sigma-Aldrich)	In vitro (1, 10 μM) In vivo (80 mg/kg)	RUNX2, OCN ↑ β-catenin, LRP5 ↑ GSK-3β ↓ ALP, OSX ↑	[101]
	Ses	Purchased (Selleck Chemicals)	In vitro (2.5, 5, 10 μM) In vivo (80, 160 mg/kg)	ALP, OCN, OPN, β-catenin ↑ TRAP, c-FOS, CTSK, NFATc1 ↓ p65, IκBα ↓	[102]
	Ses	Purchased (Sigma-Aldrich)	In vitro (0.5 μM) In vivo (80 mg/kg)	VEGF ↑ SOX9, BMP2 ↑	[103]
	Sesamolin	Purchased (Chegdu DeSiTe Biological Technology Co.)	In vitro (5, 10 μM) In vivo (5 mg/kg)	TRAP, CTSK, MMP-9, c-Fos, NFATc1 ↓ p65, IκBα, ERK, JNK, p38 ↓	[105]
Psychological disorder	(+)-Ses	Purchased (Yuanye Biotechnology)	In vivo (50 mg/kg)	5-HT, BDNF ↑ COX-2, iNOS, TNF-α, IL-1β ↓	[110]
	Ses	Purchased (Yuanye Biotechnology)	In vivo (50 mg/kg)	TNF-α, IL-6 ↓	[111]

## 5. Anticancer Effects of Lignans

Cancer is the leading cause of death in Korea. Globally, approximately 10 million patients die due to cancer annually [112]. However, conventional methods, e.g., chemotherapy or radiotherapy, carry a large patient burden and can show low efficacy. Recently, therapies directly targeting cancer-associated antigens and use of immunotherapy to modulate the immune system to avoid antigen resistance have provided new options for cancer treatment [113]. Nevertheless, the development of novel anti-tumor drugs from natural compounds with fewer adverse effects may improve cancer treatment outcomes: lignans, with excellent anti-inflammatory effects, may be one such category of cancer drugs derived from natural compounds.

### 5.1. Breast Cancer

Approximately 15% of breast cancer patients died within five years regardless of therapeutic treatment, according to the statistics from American Cancer Society [114]. Some of the patients suffer from hormone receptor-positive breast cancer receiving drugs that block estrogen receptors. Since dietary lignans mimic estrogen and are active in estrogen-receptor-mediated signaling pathways, many studies have examined the effects of dietary lignan consumption on breast cancer [115]. Interestingly however, Seco isolated from flax seeds downregulates the proliferation of MCF-7 cancer cells by exerting inhibitory activity on ERα [116]. In addition, SDG, the glycosylated form of Seco, synergistically supported the anti-cancer effect of doxorubicin, a conventional chemotherapeutic agent [117]. High-dose SDG treatment increased serum END and ENL concentrations, which led to downregulation of the transcriptive activity of NF-κB, inhibiting the survival and proliferation of MDA-MB-231 and MCF-7 cells in mice [118]. Trans-(-)-kusunokinin is a lignan compound derived from Piper nigrum known to target various breast cancer cell lines [119]. Interestingly, it exerted anti-tumor effects on breast cancer cell lines by cell-cycle arrest and induction of apoptosis [120].

### 5.2. Colorectal Cancer

Colorectal cancer is the third major cause of death globally in both men and women [114]. The lethality of colorectal cancer arises from its propensity to metastasize to the liver [121]. However, dietary lignans isolated from flax and sesame seeds may act as inhibitors of such metastasis. The postulated mechanism is based on the regulatory role of Lar, Med, and Pino on the cellular autophagy system that is directly linked with cancer survival and metastasis [122]. Dietary lignans are able to block UNC-51-like autophagy activating kinase 1/2 (ULK1/2), key regulators of metastatic colorectal cancer [123]. Surprisingly, Yefei Huang et al. discovered that Ses can inhibit metastasis of colorectal cancer by blocking angiogenesis [124]. After conducting both an in vivo angiogenesis assay and an in vitro tube-formation assay, the researchers confirmed that hypoxia-induced angiogenesis was suppressed by Ses treatment, which inhibited the NF-κB/hypoxia-inducible factor 1-alpha (HIF-1α)/VEGFA axis. In addition, SDG induced apoptosis of SW480 human colorectal cancer cells by upregulating apoptosis-inducing factor (AIF) and caspase 3 gene expression [125]. Consistent with these results, Tuo Chen et al. found that SDG mediated caspase 1 activation to induce pyroptosis of HCT116 colorectal cancer cells [126].

### 5.3. Prostate Cancer

The second leading cause of death among men worldwide is prostate cancer, with a mortality rate of 11% annually [114]. Androgens and their receptors play crucial roles in the progression of prostate cancer. The expression of androgen receptors is mainly regulated by the NF-κB pathway, and elevated NF-κB expression is detected in most prostate cancer patients [127]. Certain phytoestrogens are excellent modulators of inflammation, inhibitors of NF-κB activity, and androgen receptor (AR) antagonists, blocking the effects of androgens, such as testosterone or dihydrotestosterone (DHT) [72]. The Syr lignan, which suppresses the NF-κB pathway, has a greater binding affinity for Ars according to molecular dynamics studies [77,128]. In addition, Syr may target mutant Ars found in castration-resistant prostate cancer (CRPC), which often shows resistance to androgen-deprivation therapy [129]. 

### 5.4. Other Cancers

Lignans also have therapeutic effects in other cancers, notably with regard to the programmed-cell death pathway or phosphoinositide 3-kinase (PI3K)/Akt-mediated cancer cell proliferation (Table 2). For instance, Ses exerted anti-tumor effects in the EL4 mouse lymphoma cell line [130]. The Ses lignan induced the intrinsic apoptosis pathway by deactivating Bcl-2 and initiated pyroptosis by increasing expression of IL-1β. The flax seed lignan metabolite ENL inhibited proliferation of the KG-1 and Monomac-1 acute myeloid leukemia cell lines [131]; 100 μM of ENL triggered DNA fragmentation and apoptosis in both cell lines.

## 6. Conclusions and Perspectives

In this review, we aimed to describe the recent research addressing the antioxidant, anti-inflammatory, anti-menopause, and anti-cancer effects of lignans. Previously published data indicated the antioxidant properties of lignans are mainly related to the regulation of radical scavenging enzymes, e.g., SOD and CAT. Administration of SDG alleviated liver and kidney damage by enhancing the expression of SOD and CAT. Moreover, upregulation of the Nrf2-ARE axis, which leads to expression of HO-1, has been reported following SDG or Lari administration. Especially, Lari showed remarkable inhibition capability of cellular ROS generation comparable to gallic acid [58]. The anti-inflammatory effects of lignans are the result of NF-κB and MAPK pathway inhibition. Induction of pro-inflammatory cytokines, such as TNF-α, IL-6, IL-8, and IL-1β, by activation of the p50/p65 and c-Jun/c-Fos transcription factors was downregulated by lignans in various in vitro and in vivo models. Since the molecular mechanism of the lignans on inflammatory response has been elucidated in detail, further applicability to chronic inflammatory disease was addressed in this review.

Lignans have been suggested as possible therapeutic agents to reduce symptoms related to menopause. Many researchers have emphasized the value of lignans as a substitute for estrogen. However, recent studies showed that anti-inflammatory properties of lignans mediated by deactivation of the NF-κB and AP-1 signaling pathways can also alleviate severe bone diseases caused by estrogen deficiency in female. Moreover, their role as eNOS activators suggests the potential of lignans against cardiovascular diseases often seen in menopausal women. Lignans also alleviate neuroinflammation, reducing brain damage by preserving BDNF expression and modulating neurotransmitter levels. Surprisingly, some of the lignans showed impressive efficacy on postmenopausal symptoms compared to extrinsic hormones such as parathyroid hormone or estrogen [88,103]. It was confirmed that TNF-α expression was more strongly suppressed when treated with lignans derived from *Sambucus Willaimsii* compared to parathyroid hormone treatment. Osteoporosis alleviating factors were also upregulated by Ses treatment to the same extent as estrogen treatment. These findings indicate that there is the specific curative mechanism of lignans in the treatment of postmenopausal disorders and these compounds can substitute hormonal therapies. Lignans also target cancer cells by suppressing the NF-κB pathway and modulating apoptotic pathways. Interestingly, although breast cancer patients are counselled to avoid consuming dietary lignans, as they upregulate the estrogen receptor-mediated pathway, some lignans, e.g., SDG or trans-(-)-kusunokinin, have antitumor effects on breast cancer cells by inhibiting Akt, Cyclin D1, and CDK to reduce proliferation. Their role as inflammation modulators also emphasizes the function of lignans in other cancers like colorectal, prostate, and blood cancers.

Although this review does not cover entire studies of lignans on inflammatory diseases, we hope that the review presents the direction of therapeutic approaches on lignans against several major diseases, such as cancerous, inflammatory, and cardiovascular diseases. Lignans are typically formulated as functional food items and administered orally. Few studies have been conducted to develop drugs based upon lignans and to design specific carriers to increase the delivery efficiency of lignans, possibly because of the uncertainty regarding the efficacies of lignan polymers in various chronic diseases. However, recent studies have shown that parent lignans also exert numerous curative effects on human chronic diseases, not just the active END or ENL metabolites. Therefore, more efforts should be made to develop lignan-based drugs against chronic inflammatory diseases. Based on previous studies with differing target concentrations and lignans, a specific formulation should be determined for a pharmaceutical preparation containing various lignan compounds. We expect that lignans will soon not only be used as functional foods, but also in the pharmaceutical industry.

## Figures and Tables

**Figure 1 ijms-23-15482-f001:**
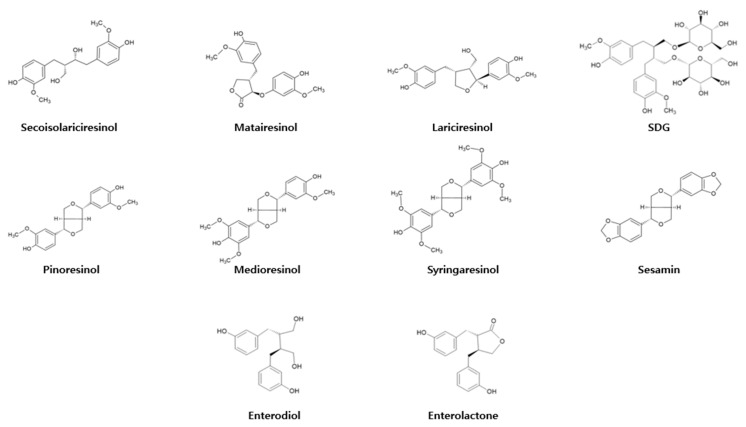
Chemical structures of lignans and their metabolites (END and ENL).

**Figure 2 ijms-23-15482-f002:**
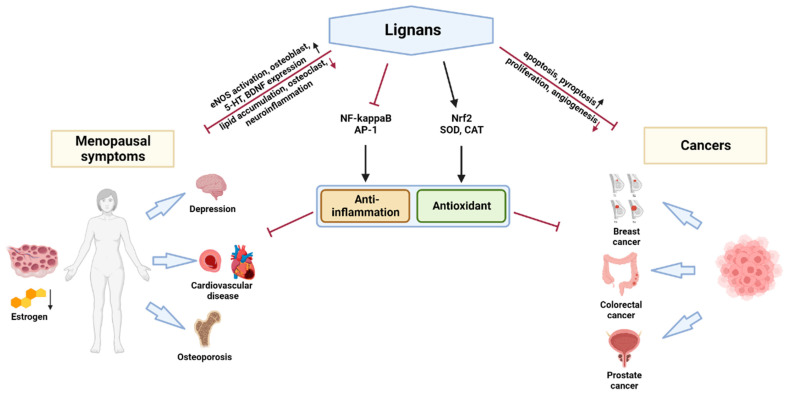
Schematic analysis of the therapeutic properties of lignans.

**Figure 3 ijms-23-15482-f003:**
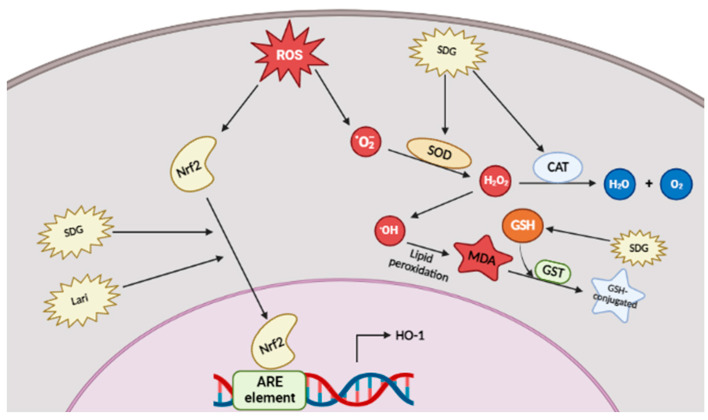
Antioxidant properties of lignans. Intracellular radical scavenging process can be stimulated by SDG by enhancing the expression of SOD and CAT as well as MDA reduction. Lari and SDG target Nrf2-mediated expression of HO-1.

**Figure 4 ijms-23-15482-f004:**
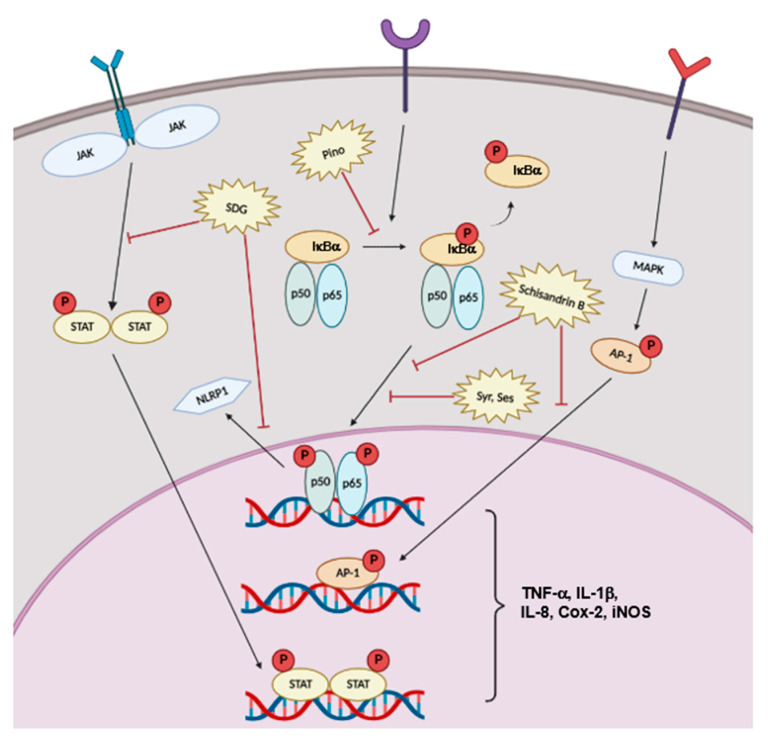
Anti-inflammatory properties of lignans. Various lignans block the expression of pro-inflammatory cytokines via downregulating JAK/STAT, NF-κB, and AP-1 pathways.

**Table 2 ijms-23-15482-t002:** Summary of the results of published pharmacological experiments conducted to evaluate the effects of lignans in cancers.

Disease	Lignan	Source	Test Type and Dose	Molecular Mechanism	Ref.
Breast	Seco	*L. usitatissimum*	In vitro	PARP cleavage ↑ Erα ↓	[116]
	SDG	*L. usitatissimum*	In vitro (1, 10 μM) In vivo (25, 74 mg/kg)	END, ENL ↑ p65 ↓	[118]
	(-)-Kusunokinin	*P. nigrum*	In vitro (1.6, 3.2, 6.4 μM)	p53, p21 ↑ Bcl-2 ↓ Bax, Cytochrome c, caspase 7/8 ↑	[120]
Colorectal cancer	Mat, Pino, Lari, Seco, Medi	Structures from pubchem database	In silico	ULK1/2 ↓	[123]
	Ses	Purchased (Sellek Chemicals)	In vitro (10 μM)	IκBα, p65 ↓ HIF-1α ↓ VEGFA ↓	[124]
	SDG	Purchased (Sigma-Aldrich)	In vitro (100, 150 μM)	AIF, caspase 3 ↑	[125]
	SDG	Purchased (MedChemExpress)	In vitro (50 μM)	GSDMD, caspase-1, cytochrome c, Bax ↑ PI3K, Akt ↓	[126]
Prostate cancer	Syr	Structure from pubchem database	In silico	AR ↓	[128]
T cell Lymphoma	Ses	Purchased (TargetMol)	In vivo (10 mg/kg) In vitro (10, 20, 40 μM)	Cyclin D1 ↓ caspase 3, Bax ↑ Bcl-2 ↓ caspase 1, NLPR3 ↑ Atg5, LC3 II/I ↑ p62 ↓	[130]
Acute Myeloid Leukemia	ENL	Purchased (Sigma-Aldrich)	In vitro (40, 100 μM)	Cytochrome c, PARP, Bax, caspase 3/9 ↑ Bcl-2 ↓	[131]

## Data Availability

The data are contained within the article.

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
