# Peer review of "Antioxidant, Anti-Inflammatory, Anti-Menopausal, and Anti-Cancer Effects of Lignans and Their Metabolites"

_ijms, 2022, doi:10.3390/ijms232415482_

Round 1

Reviewer 1 Report

This review provides an interesting summary of recent studies about curative activity of different lignans, including antioxidative, anti-inflammatory, anti-menopause, and anti-cancer actions. Nevertheless, a few issues are needed to be clarified.

1. Please check the formatting of the manuscript. There are several cases when subscripted/superscripted text does not appear correctly.

2. The arrows in table 1 are in a jumble and not conducive to easily reading. Better arrangement needs to be explored.

3. Two figures in the paper are not quite clear, and it is better to move figure 2 to introduction.  

Reviewer 2 Report

 Comments:

1.      Abstract is too short; it needs to write more about Lignans and Their Metabolites.

2.      Correct the sentence In line 19, by removing the “an” and add “s” in agent. Correct is…… values as effective anti-inflammatory agents.

3.      Correct the statement and insert the reference:

 Increased cytokine production induced by NF-κB pathway activation in various organs may eventually lead to severe cardiovascular disease (including atherosclerosis), osteoporosis, and/or psychologic disorders such as depression.

4.      The 3-d sketch of the Chemical structures of lignans could be better. It needs to re-draw all the chemical structures to improve the quality.

5.      The information about NF-κB is very less in the introduction. There should be more addition about the role of NF-κB and its pathways in the text (https://www.eurekaselect.com/article/111395  10.2174/0929867327666201111142307, https://www.biotech-asia.org/vol16no4/exploring-inhibitory-mechanisms-of-green-tea-catechins-as-inhibitors-of-a-cancer-therapeutic-target-nuclear-factor-%CE%BAb-nf-%CE%BAb/)

6.      Considering that the present manuscript is not a systematic review. I would suggest to authors depict more figures on how lignans and their metabolites exert anti-inflammatory effects by modulating both the NF-κB and MAPK pathways in section 3 Anti-Inflammatory Properties of Lignans.

7.      Figure 2 needs a lot of improvement. First, the quality of Figure 2 is inferior. I suggest using software like Bio render to depict the pathways. It could be improved by adding novelty and originality, focusing on novel compounds, etc.

Reviewer 3 Report

Dear Authors,

The article contains comprehensive knowledge about the antioxidants and anti-inflammatory potentials in various chronic diseases. I felt the authors should have focused more on  Lignans and their Metabolites in comparison with natural antioxidants like curcumin, stilbenes, etc

The rest of the information gathered by the authors is sufficient. All the best.

Author Response

Reviewer 3

The article contains comprehensive knowledge about the antioxidants and anti-inflammatory potentials in various chronic diseases.

  1. I felt the authors should have focused more on Lignans and their Metabolites in comparison with natural antioxidants like curcumin, stilbenes, etc

Answer:

àThank you for your comments. We find that some reference articles utilize other compounds (ex) gallic acid, estrogen) as positive control to compare the effects of lignans. Therefore, we revised the 6. Conclusion section as below (see L346-347 and L361-369).

***Especially, Lari showed remarkable inhibition capability of cellular ROS generation comparable to gallic acid [40].

***Surprisingly, some of the lignans showed impressive efficacy on postmenopausal symptoms compared to extrinsic hormones such as parathyroid hormone or estrogen [70,85]. It was confirmed that TNF-a expression was more strongly suppressed when treated with lignans derived from Sambucus Willaimsii compared to parathyroid hormone treatment. Osteoporosis alleviating factors were also upregulated by Ses treatment to the same extent of estrogen treatment. These findings give us messages that there is the specific curative mechanism of lignans in the treatment of postmenopausal disorders and these compounds can substitute hormonal therapies.

Reviewer 4 Report

The authors carry out a non-systematic review of the effect of lignans on various pathologies where inflammation plays a key role. They make a broad introduction of these pathologies and later comment on the effects of these phytoestrogens. They present a final figure ( Figure 2 ) that is very explanatory about the effects of the lignams that are complemented by previous tables. The language is very motley making the review difficult to read. It would probably be interesting if the figure appeared at the beginning of the review and use it as a basis for further discussion. The limitations of the review should be stated as well as the methodology used.

Author Response

Reviewer 4

The authors carry out a non-systematic review of the effect of lignans on various pathologies where inflammation plays a key role. They make a broad introduction of these pathologies and later comment on the effects of these phytoestrogens. They present a final figure ( Figure 2 ) that is very explanatory about the effects of the lignans that are complemented by previous tables.

  1. The language is very motley making the review difficult to read. It would probably be interesting if the figure appeared at the beginning of the review and use it as a basis for further discussion.

Answer:

à Thank you for your comments. We’ve simplified the figure 2. And it has been moved from the figure 2 to introduction (see p3).

  1. The limitations of the review should be stated as well as the methodology used

Answer:

àThank you for your comments. We think over the limitations of the review. We only focused on recent research of testing lignan efficacy not covering entire studies. However, we hope that this review will give the direction toward future lignan studies, so we stated as below (see: 376-379).

“Although this review does not cover entire studies of lignans on inflammatory diseases, we hope that the review presents the direction of therapeutic approaches on lignans against several major diseases such as cancerous, inflammatory, and cardiovascular diseases.”

àThank you for your comments. We also added methodology of the review in introduction part as below (see L77-80)

For this, we referred to the experimental papers published within five years that show efficacy of lignans. We set various lignan compounds including the seven classes and their metabolites as key words and searched studies in Pubmed and Embase.

Round 2

Reviewer 2 Report

Although authors have improved the manuscript significantly, still  there are some flaws which needs to be fixed before the publication of this manuscript.

1.      Resolution of Figure 2. Schematic analysis of the therapeutic properties of lignans, is very dull which further need to improve.

2.      Authors should write the figure legends in figure 3 and figure 4.

3.      In L-52, This renders to release NF-kB subunits and trigger….correct it triggers.

Author Response

Reviewer #2

Although authors have improved the manuscript significantly, still  there are some flaws which needs to be fixed before the publication of this manuscript.

  1. Resolution of Figure 2. Schematic analysis of the therapeutic properties of lignans, is very dull which further need to improve.

Answer: We have revised it to improve it (see Fig. 2).

  1. Authors should write the figure legends in figure 3 and figure 4.

Answer: We have added the figure legends in Fig. 3 and 4, according to this comment (see L114-116 and 156-157).

  1. In L-52, This renders to release NF-kB subunits and trigger….correct it triggers.

Answer: Yes, we have added it to trigger (see L53).

Reviewer 4 Report

The responses of the authors are correct. The paper is better that the previous version

Author Response

Thanks very much for your reviewing.